# OpenReview Should be Protected and Leveraged as a Community Asset for Research in the Era of LLMs

## Abstract

In the era of large language models (LLMs), high-quality, domain-rich, and continuously evolving datasets capturing expert-level knowledge, core human values, and reasoning are increasingly valuable. **This position paper argues that OpenReview — the continually evolving repository of peer reviews, author rebuttals, meta-reviews, and decision outcomes — should be leveraged more broadly as a core *community asset* for advancing research in the era of LLMs.** We highlight three promising areas in which OpenReview can uniquely contribute: enhancing the quality, scalability, and accountability of peer review processes; enabling meaningful, open-ended benchmarks rooted in genuine expert deliberation; and supporting alignment research through real-world interactions reflecting expert assessment, intentions, and scientific values. To better realize these opportunities, we suggest the community collaboratively explore standardized benchmarks and usage guidelines around OpenReview, inviting broader dialogue on responsible data use, ethical considerations, and collective stewardship.

## 1 Introduction

The past years have witnessed an extraordinary shift in the role of data within machine learning [1, 2], especially with the recent advances of large language models (LLMs) [3–5], which have progressed from task-specific tools to general-purpose reasoning engines [6–8]. As their capabilities expand across domains, the role of data for training, evaluation, and alignment becomes even more important [9–12]. The current wave of LLM development increasingly depends on high-quality, human-centered feedback [13–17], not only for fine-tuning and instruction adherence, but also for assessing model behavior, identifying failure modes, and aligning outputs with human expectations [18–21]. Yet many of the datasets used for these purposes remain limited in coverage [22], synthetic in composition [23, 24], or static in structure [15]. As a result, they often fail to capture the complexity, disagreement, and subtle reasoning that characterize authentic human judgment [25–28].

At the same time, the powerful capabilities of LLMs are beginning to reshape scientific workflows themselves [29–33]. Tools based on LLMs such as ChatGPT are making research communication, including literature reviews and even paper writing, more accessible [34–37], hence accelerating scientific output and contributing to a significant rise in the volume of submissions to top conferences. Such a transformation has intensified pressure on the peer review system [38, 39]. Conferences now receive more than 10 thousands of submissions per cycle, and the human effort required to maintain high-quality, fair, and constructive reviewing has become difficult to sustain. Given such high pressure, the need for scalable assistance tools, better evaluation data, and models that can understand or generate scholarly critique has grown [39–41]. However, large-scale, systematic exploration regarding both the datasets and methodologies that enable LLMs to capture the richness of peer review interactions is still missing [42–44].

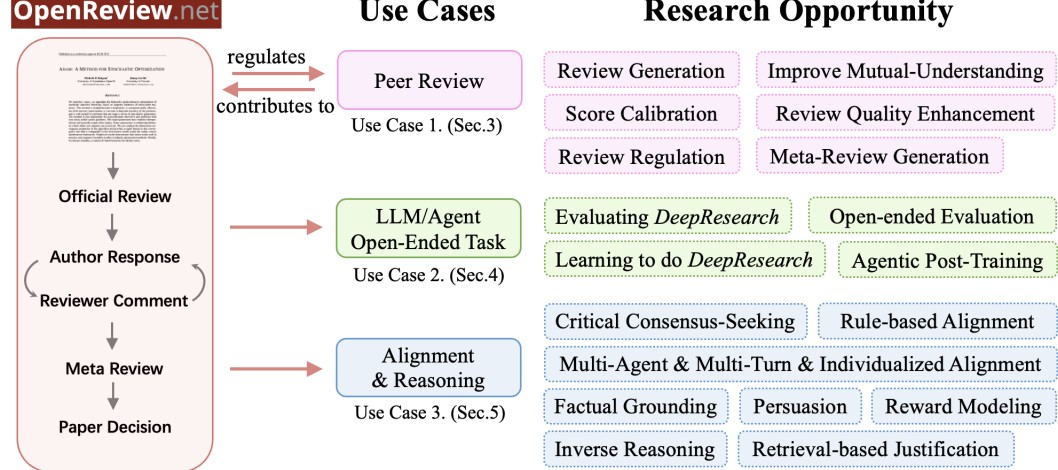

Figure 1: **Left**: an overview of the OpenReview data generation process; **mid**: this position paper argues OpenReview supports three main valuable applications — regulating peer review, empowering LLM and Agentic open-ended task research, and post-training for alignment and reasoning; **right**: highlighted research opportunities around those use cases.

OpenReview[1][45], the public review platform widely used by conferences such as ICLR, NeurIPS, and others, offers a unique opportunity to meet the needs of both sides. Contributed by the community and continually expanding over time, OpenReview hosts large-scale, structured records of scientific discussion, typically including paper submissions, reviewer assessments, author rebuttals, meta-reviews, and final decisions. These interactions span multiple rounds and involve diverse expert perspectives, making OpenReview an invaluable living dataset grounded in real-world scientific research deliberation. And has the potential to enrich both data-centric LLM research and assist the peer review system.

**This position paper argues that OpenReview should be leveraged more broadly as a core community asset for advancing research in the era of LLMs.** We elaborate on three areas where this dataset can provide immediate value:

1. **A data-driven approach to improve the quality and scalability of peer review.** OpenReview provides a unique source of structured, expert-generated assessments that can be used to train machine learning models to analyze and support the peer review process. Machine learning models, including the state-of-the-art general purpose language models [46–49], may learn to assist reviewers in drafting constructive feedback, calibrating scores, and identifying argumentative gaps, as well as summarizing responses, checking code, or detecting unhelpful language. In the face of rising submission volumes and reviewer fatigue, such tools could support more consistent, fair, and informative evaluations. Equally important, improving and regularizing the review process is a **prerequisite** for sustaining the long-term development of LLM-based systems that depend on high-quality expert feedback [50].

2. **Providing expert-generated benchmarks for LLM open-ended task evaluation and post-training.** Open-ended tasks such as academic writing, research evaluation, persuasion, or summarization are increasingly recognized as central to the development of general-purpose AI systems and the path toward superintelligence [51]. However, both training and evaluating models on such tasks remain challenging due to the open-ended nature and the lack of scalable, high-quality human feedback [8]. To this end, OpenReview offers a unique, high-quality resource: it contains expert-curated, multi-dimensional evaluations of research contributions grounded in real-world scientific progress. Its diverse content enables the design of benchmarks for open-ended tasks such as writing [52], research evaluation [40], persuasion [53–55], and summarization [16, 56], providing valuable data for both open-ended LLM and agentic post-training and evaluation [57].

3. **Supporting multi-dimensional alignment and reasoning research through scientific writing and discussion.** Existing benchmarks for alignment and reasoning often rely on static, synthetic, or crowd-sourced datasets that lack the depth and nuance of real expert deliberation [15, 58–62].

---

[1]https://openreview.net/

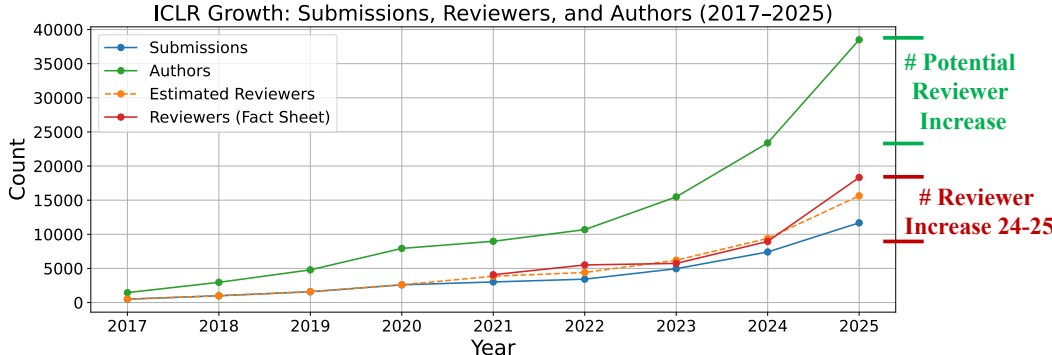

Figure 2: Growth trends at ICLR (2017–2025) in submissions, authors, and reviewers. While the number of reviewers has increased over time, it has not kept pace with the growth in submissions and authors, indicating a growing strain on the peer review process. The reviewer number estimation is calculated according to the number of submissions, the total number of reviews received, and the average reviewer workload of 3 per reviewer.

In contrast, OpenReview offers a setting that inherently involves alignment and reasoning through evidence-based debate, disagreement, revision, and consensus building. This setting enables rich evaluation tasks such as score justification via retrieval-based reasoning [63–66] and decision prediction grounded in free-form critique [67]. These tasks can serve as realistic testbeds for assessing how well LLMs can interpret, reason about, and align with expert judgments in the scientific research domain. Moreover, the dialogic nature of OpenReview — spanning rebuttals, conflicting views, and negotiated outcomes — offers a unique opportunity to study value pluralism, debate-style alignment in the wild [53, 68–71].

To help realize the potential in and beyond those outlined use cases, we propose initial directions for community-driven benchmark development and responsible data stewardship. Finally, we reflect on alternative perspectives, aiming to spark productive dialogue on the challenges and risks of leveraging the OpenReview as a core community asset.

## 2 The State of OpenReview Now: Scale, Opportunity, and Emerging Risks

This section examines the OpenReview platform through three perspectives. We begin with a statistical overview of its scale and evolution, using ICLR as a case study. We then highlight its value as a unique community-curated dataset for machine learning research, before turning to the structural risks that threaten its long-term quality and integrity.

### 2.1 The Scale and Structure of Conference Data on OpenReview

OpenReview provides a centralized platform for peer review and community discussion at major machine learning conferences, including ICLR, NeurIPS, and others. It preserves structured records of submissions, reviews, rebuttals, and decisions, creating a longitudinal archive of real-world expert deliberation under consistent guidelines.

To illustrate the scale of this platform, we focus on ICLR as a representative case. From 2017 to 2025, the number of submissions grew from fewer than 500 to over 11,600 annually. The corresponding number of authors increased from about 1,500 to 38,500, and the estimated number of reviewers rose from under 1,000 to more than 18,300. Each submission typically receives three or more expert reviews, resulting in tens of thousands of reviewer–author interactions each year. Figure 2 shows this growth trajectory in authorship, reviewing, and participation.[2]

### 2.2 A Rapidly Growing Community Asset for Learning

Beyond its scale, OpenReview is distinguished by its unique data quality. Unlike synthetic or crowd-sourced datasets, it captures expert-authored evaluations tied to real submissions and decisions,

---

[2]Data Source: ICLR 2021-2025 Fact Sheet [72–76], PaperCopilot [77].

grounded in shared scientific norms. Each paper serves as a self-contained research scenario, typically accompanied by multiple reviews, optional rebuttals, meta-reviews, and final outcomes.

Between 2017 and 2025, ICLR alone contributed over 36,000 such interaction threads, spanning both accepted and rejected submissions. These interactions provide rich examples of open-ended scientific exploration efforts. They illustrate how researchers conduct and evaluate solutions to open questions, respond to disagreement, clarify claims, and finally construct consensus, making them highly suitable for training and evaluating LLMs on scientific reasoning, argumentation, and alignment.

Moreover, OpenReview is a continuously evolving dataset. Each year brings new topics, new papers, and new debates, reflecting both the state of research and the shifting consensus of the community. This ongoing refresh ensures its competence as a benchmark for real-world LLM deployment. In Table 1, we compare relevant tasks in the LLM post-training community to demonstrate the general potential of the OpenReview dataset.

Table 1: Comparing datasets related to OpenReview. We will elaborate how to leverage OpenReview beyond those tasks in Sec.3-5.

| Dataset | Task | Size | Expert | Updates | OpenEnded |
|---------|------|------|--------|---------|-----------|
| See et al. [78] | Summarization | 310K | ✓ | × | ✓ |
| Narayan et al. [79] | Summarization | 226K | ✓ | × | ✓ |
| Yang et al. [80] | Multi-hop QA | 113K QA pairs | × | × | ✓ |
| Rajpurkar et al. [81] | Comprehension | 107K QA pairs | × | × | × |
| Fan et al. [82] | Long-form QA | 270K threads | × | × | ✓ |
| Ziegler et al. [83] | Preference Modeling | 60K comparisons | ∼ | × | ✓ |
| Bai et al. [15] | Alignment / Dialogue | 170K comparisons | ∼ | × | ✓ |
| Köpf et al. [84] | Dialogue / Alignment | 10K trees, 161K msg | ∼ | × | ✓ |
| Wang et al. [85] | Argumentation | 1K dialogues | × | × | ✓ |
| Kang et al. [56] | Review, Decision | 14.7K subs, 10.7K revs | ✓ | × | × |
| Bu et al. [86] | Aspect Rating (zh) | 46.7K reviews | ∼ | × | × |
| Purkayastha et al. [87] | Argumentation | 2.3K | ∼ | × | ✓ |
| Kennard et al. [88] | Argumentation | 506 threads | ✓ | × | × |
| Ruggeri et al. [89] | Argumentation | 41 dialogues | ✓ | × | ✓ |
| OpenReview [45] | **All above** | 36K subs, 100K+ revs | ✓ | ✓ | ✓ |

## 2.3 Quality Under Pressure — The Compounding Risk of Rapid Growth

While the growth of OpenReview presents significant opportunities, it also introduces structural risks. The rapid increase in submission volume has not been matched by a proportional increase in highly experienced reviewers. As conferences scale, an increasing fraction of reviews are written by newer or less engaged participants. This trend raises concerns about the consistency, reliability, and long-term stability of individual review signals, as well as the dataset quality and diversity [50].

More precisely, the concern is not only that current reviewers may deviate from academic standards, but that a growing number of untrained reviewers may internalize and reproduce biased practices, gradually compounding the problem across generations. If evaluations are learned by imitation, biased or inconsistent norms can propagate, leading to long-term degradation of review quality.

To formalize this concern, we present a Wright-Fisher model in Appendix A, which illustrates how misaligned reviewing behavior may propagate across generations.

> **Take Action Now.** Our analysis suggests that early intervention is critical: corrective action taken before problematic patterns become institutionalized is significantly more effective than attempting to reverse them later. Proactive steps are thus essential to preserve long-term alignment between reviewing practice and community values. **Taking action now in the early stage of the field's expansion is more effective than taking action later on when substandard review practices become the norm.**

For OpenReview to remain a robust and trustworthy resource, its quality must be actively protected. This includes better reviewer recruitment and training, as well as developing scalable, practical machine learning methods for auditing and mitigating quality drift. The data itself, while valuable, is only as good as the process that generates it.

In the following sections, we will discuss three use cases of the OpenReview dataset, starting from how to leverage the dataset to improve and regulate the peer review system, such that the long-term quality of such a community asset can be guaranteed. We then highlight the potential of leveraging such an asset in LLM post-training research, ranging from open-ended to alignment tasks.

## 3 Assisting and Protecting the Peer Review with OpenReview

### 3.1 Existing LLM-Assisted Peer Review in Conferences

In the previous section, we highlighted structural risks to the quality and stability of peer review. These concerns have not gone unnoticed. In recent years, several major machine learning conferences and publishers have begun integrating LLMs into their review workflows in response.

NeurIPS 2024 introduced a checklist assistant powered by LLMs to help authors ensure ethical and methodological compliance [90]. At ICLR 2025, a Review Feedback Agent was deployed to identify vague or unconstructive reviews and suggest targeted improvements [91]. AAAI 2026 will experiment with LLM-generated reviews and discussion summaries in the first stage of review [92]. Meanwhile, several academic publishers have begun piloting AI-assisted tools for content checking and review drafting [93–95].

While most current systems operate with limited, hand-curated inputs, OpenReview provides an ideal foundation for data-driven peer review research. In this section, we focus on concrete use cases where such data can support the review system.

### 3.2 Practices and Opportunities for Data-Driven Support with OpenReview

We organize existing literature and potential opportunities with OpenReview according to functional categories. In the following, we will use **text boxes** to highlight **future work opportunities**. The high-level motivation of those approaches is rooted in the previous success of human-centered LLM alignment research [17, 15, 16], and data-driven decision modeling and explanation [96–100].

**Principled Review Generation.** Recent work has explored OpenReview for generating *realistic* peer reviews. Yuan et al. [101] and Wu et al. [102], for example, demonstrate that fine-tuning LLMs on large-scale review corpora can lead to critiques that are more calibrated and grounded than those produced by general-purpose models. These systems can be conditioned on paper content or specific review dimensions, enabling targeted and context-aware feedback. However, most current systems are designed to *mimic* human-written reviews without deeper integration with formal reviewing guidelines or accountability structures. The challenge remains to ensure that generated reviews uphold conference standards and provide actionable feedback in line with reviewer expectations.

> **Opportunity for Future Work.** LLMs should be task-specifically aligned, calibrated when leveraged in the review process. Commercial LLMs are generally optimized for user-friendliness and helpfulness, often deviating from rigorous academic review guidelines. Future work should explore structured prompting, rubric conditioning, or alignment objectives tailored for review generation [44]. In addition, LLM-generated reviews may support pre-submission preparation [42], providing anticipatory critique to authors and supporting self-assessment before formal peer review [92].

**Review Quality Enhancement.** Another line of research focuses on the quality of peer reviews themselves. Early work, such as Kang et al. [56], proposed metrics for review helpfulness and score prediction. More recently, classifiers trained on human preferences or meta-review feedback have been developed to detect vague, biased, or uninformative reviews [103, 104]. Studies have also examined hallucination and style inconsistencies in LLM-generated reviews [105–108]. Despite these advances, challenges remain in automatically evaluating review fairness, argument soundness, or reviewer calibration.

> **Opportunity for Future Work.** Inverse analysis techniques can help detect systematic deviation from expected standards, including overconfidence, inconsistency, or subjective bias [97]. Future efforts could explore calibration, value drift detection, and provide warning signals when the value of reviews deviate significantly from guidelines [44].

**Enhancing Mutual Understanding between Reviewers and Authors.** While much of the focus has been on generating or evaluating individual reviews, peer review is ultimately a dialogue. The rebuttal phase plays a crucial role in bridging perspectives between authors and reviewers. Recent datasets such as DISAPERE [109], Jiu-Jitsu [110], and ContraSciView [111] support tasks such as rebuttal generation, stance classification, and discourse structure prediction, highlighting the interactional nature of review.

> **Opportunity for Future Work.** LLMs can serve as mediators to enhance communication in the rebuttal process. For authors, they may clarify reviewer concerns, highlight overlooked critiques, and assist in crafting respectful and persuasive responses for effective communication. For reviewers, they may help interpret rebuttals and assess whether key feedback has been adequately addressed, and effectively stimulate the discussions.

**Consistency and Calibration**. Efforts to correct score inconsistency across reviewers have drawn on reviewer calibration and normalization techniques. For instance, Xu et al. [112] models reviewer-specific scoring functions and applies monotonic transformations to improve comparability. These methods aim to recover more faithful rankings than simple score averaging. Nonetheless, current approaches often lack interpretability or real-time applicability. There is limited support for helping reviewers understand their own biases or dynamically recalibrate scores based on peer context.

> **Opportunity for Future Work.** More importantly and effectively, efforts could be made to use LLM-based systems to assist reviewers in providing consistent and calibrated feedback, including providing comparative context and relevant arguments drawn from reviewer cohorts [42]. Technically, this may involve retrieval-based justification of scores and decision explanation [63, 64, 113, 114], or in-context learning reference sample selection [115, 116].

**Meta-Review Generation.** Finally, meta-review generation has become a growing area of interest, with benchmarks such as PeerSum [104], ORSUM [117], and MOPRD [118] targeting summarizing and concluding from multiple reviews and rebuttals. These systems must integrate conflicting reviewer perspectives, identify dominant themes, and represent area chair judgment with fidelity. Still, current general-purpose LLMs may fail to capture the nuanced reasoning behind disagreements or the weight assigned to various critiques. There is also growing concern about the potential mismatch between generated meta-reviews and actual reviewer consensus [39].

> **Opportunity for Future Work.** Improved modeling of review disagreement and viewpoint clustering [68, 70] could enable more reliable meta-review generation. Future systems may incorporate hybrid workflows where LLMs co-author drafts with area chairs, flag unresolved conflicts, or highlight potential biases (e.g., delayed or biased feedback, ungrounded critiques) throughout the discussion period to support better decision making.

# 4 OpenReview for Open-Ended Task Evaluation and Post-Training

## 4.1 Challenges for Open-Ended LLM and Agentic Tasks

Recent progress in LLMs has enabled systems that attempt to perform complex, multi-step, and high-level tasks, often referred to as *open-ended* or *agentic* tasks [119, 120]. These tasks are characterized by the absence of a single correct answer, dependence on context, and the need for judgment, reasoning, and creativity [52]. Examples include research paper writing, paper reviewing, persuasive argumentation, hypothesis refinement, and code-based experimentation [121, 122]. Open-ended tasks are defined not by accuracy or success alone, but by depth, coherence, exploration, and alignment with human values and intentions.

This task category has received increasing attention with the rise of agent-based systems such as DeepResearch, DeepSearch, and AutoDev, which aim to position LLMs as autonomous research assistants capable of conducting literature reviews, designing experiments, debugging code, and evaluating progress [8, 123, 124]. However, a major bottleneck in building and benchmarking such systems lies in the lack of scalable, high-quality supervision. It remains difficult to evaluate whether a model has conducted a "good" literature review or proposed a "promising" research idea, particularly when using crowd-sourcing judgment [52, 121].

Scientific research, especially in machine learning, is itself an open-ended task. The process involves formulating problems, iterating on designs, running experiments, interpreting results, engaging with criticism, and ultimately persuading a community of experts. Despite this, most benchmarks for evaluating LLMs remain synthetic, short-form, or not scalable, offering little insight into how models would perform under the standards and expectations of actual research communities [121, 122].

This gap motivates our focus on OpenReview as a valuable, underutilized resource. The rich interactions on OpenReview suggest two distinct forms of supervision that are particularly suited for open-ended task development.

### 4.2  Two Potential Supervision Streams from OpenReview

**Scientific Demonstrations: Training LLMs to Do Research.**  Each submitted paper on OpenReview can be viewed as a real-world demonstration of open-ended problem-solving. Papers span a wide range of topics and contain full narratives of how authors design and communicate their contributions. This includes technical framing, literature positioning, experimental results, and claim justification. In aggregate, these documents offer structured demonstrations of how research is conceived, executed, and defended [56].

Such examples can be used to train LLMs to follow the cognitive workflow of scientific research. In particular, they can support training for complex capabilities such as multi-stage planning, tool use, fact retrieval, and hypothesis revision. These capabilities align closely with the demands of emerging agentic LLM frameworks [125]. While systems like ChatDev simulate these workflows [126], few are grounded in real, high-quality demonstrations of how experts actually perform these tasks — OpenReview offers a scalable source of such supervision.

> **Opportunity for Future Work.** OpenReview's corpus of research demonstrations can support training of LLM agents to perform multi-step scientific reasoning under real-world constraints. Future work may consider enhancing the agentic research capabilities of LLMs [8] using expert scientific research demonstrations.

**Structured Evaluations: Training LLMs to Evaluate Research.**  In addition to research demonstrations, OpenReview also contains detailed records of how experts evaluate open-ended research work. Reviews provide constructive feedback, numerical scores, and qualitative assessments, while meta-reviews offer consensus summaries and rationales for decisions. Author responses further enrich the discourse, revealing how researchers engage with critiques and attempt to clarify or defend their contributions. These dual supervision signals are particularly valuable for developing and evaluating general-purpose models intended to reason about, participate in, and evaluate complex open-ended tasks given scientific standards. By learning from those debates, LLMs have the potential to gain the capability to comprehensively evaluate open-ended research.

> **Opportunity for Future Work.** OpenReview's review traces can serve as supervision for LLM-based evaluators trained to judge open-ended research quality. These include automated meta-reviews, rebuttal critiques, and scoring models aligned with human preferences. With those feedback-rich reward models for open-ended tasks, future work can better anchor and be optimized for open-ended research.

## 5  OpenReview as High-Quality Dataset for Alignment and Reasoning

### 5.1  Challenges for Alignment and Reasoning Supervision

**Alignment through Consensus-Seeking.** Alignment research seeks to ensure that AI systems act according to human values, preferences, satisfy human intentions, and guarantee safety [22, 127]. Recent advances in reinforcement learning from human feedback (RLHF) [14–16, 128–134] have contributed to the success of LLMs in conversational systems [14]. Yet many of these advances rely on limited forms of supervision: crowd-sourcing annotations [26], synthetic preferences [59], or binary votes [135]. These sources often fail to capture the complexity, depth, and disagreement inherent in the multi-perspective and deliberative consensus-seeking processes of experts [129, 131, 136].

**Reasoning beyond Binary Tasks.** On the other hand, reasoning ability has become the core in enhancing the models' performance on more general assistant tasks [137]. Contributing to such progress, datasets such as GSM8K [138], MATH [139], and HotpotQA [80] have driven rapid progress in mathematical and multi-hop reasoning; techniques like (long-)chain-of-thought [66, 65, 140, 7] and retrieval-augmented methods [63, 64, 113] have significantly improved model performance on these structured tasks. However, many of these benchmarks are now nearly saturated by frontier models [141], focus on binary and verifiable tasks, and they predominantly focus on final answer correctness rather than the quality or interpretability of reasoning processes [142, 143].

More fundamentally, current reasoning tasks are often limited by narrow scope, synthetic formulation, or rigid answer structures [144–146]. Most define a single ground-truth answer, which precludes exploration of ambiguity, disagreement, or multi-agent deliberation, which are central to human reasoning, but effective in eliciting deep thinking behaviors [147, 137]. Although emerging datasets in argumentative reasoning, such as DebateSum [148] and OpenDebateEvidence [149], have expanded the scope of evaluation to include summarization and contested claims, these resources remain rare and are typically not grounded in scientific domain expert-generated contexts.

## 5.2 Opportunities with the OpenReview Dataset

In contrast, OpenReview offers a fundamentally different alignment and reasoning testbed. The peer review process inherently involves dialogue in which multiple parties express values, critique reasoning, and negotiate consensus. More importantly, those dialogues, in principle, should be *objective*, centered around guidelines, and grounded in verifiable facts. Unlike existing alignment datasets, which are largely *subjective*, static, and one-shot, OpenReview captures multi-round, multi-agent interactions grounded in real, highly verifiable, and reproducible consequences. This makes it a uniquely rich environment for alignment and reasoning research.

**Learning to Reason from Expert Disagreement and Justification** With OpenReview, models can be trained to infer about the logic behind review scores by learning from rationales, a form of inverse reasoning that links decisions to supporting arguments and context. The reviews themselves often present well-defined reasoning chains that connect experimental design, observed outcomes, and stated conclusions. These examples allow LLMs to practice multi-step reasoning, assess methodological soundness, and trace causal explanations. Moreover, OpenReview enables modeling how reasoning develops through multiple rounds of interaction: authors respond to critiques, reviewers clarify concerns, and final evaluations synthesize evolving viewpoints, offering a natural setting for studying the rationale behind reasoning over time.

> **Opportunity for Future Work.** Using OpenReview in future works, it's possible to improve models' reasoning abilities by justifying numerical assessments, verifying scientific claims through factual evidence, and adapting reasoning across multi-stage interactions.

**Learning to Critically Align with Individual Preferences** OpenReview provides a valuable foundation for developing alignment strategies that move beyond superficial agreement. Unlike many alignment datasets that prioritize helpfulness or user-pleasing responses, peer review process demands that feedback remain grounded in correctness, rationality, and align with review guidelines, when given diverse research contexts.

Each reviewer expresses their judgment through both numerical scores and detailed commentary, guided by criteria such as novelty, technical soundness, and significance. These preferences are dynamic and can shift in response to rebuttals and clarifications, offering supervision signals for modeling alignment as a contextual and adaptive process.

> **Opportunity for Future Work.** OpenReview enables the alignment of LLMs to offer diverse, constructive, evidence-based critique. Rather than merely affirming user input, models can learn to respectfully challenge flawed claims, explain counterarguments, and justify disagreement. This supports the development of alignment systems that emphasize factual grounding, logical reasoning, and responsible communication.

# 6 A Call to Create Standardized Benchmarks Based on OpenReview

In this section, we turn to the foundational infrastructure required to realize their full potential: standardized benchmarks and responsible community stewardship.

Despite its scale and richness, OpenReview remains underutilized as a research asset, primarily due to the lack of well-defined tasks and shared evaluation pipelines. To address this gap, we propose that the community collaboratively develop benchmarks in critical areas such as review quality assessment, rebuttal generation, argument grounding, and meta-review summarization. And deploy developed methods to intervene in the peer review system and improve its quality as soon as possible. These tasks are directly tied to the health of the peer review process and, by extension, the integrity of the dataset itself.

In parallel, more general tasks—including reviewer score prediction, open-ended task evaluation, post-training, alignment, and reasoning enhancement—can also be standardized to support long-term research. While these areas are essential to the development of LLMs, their delayed investigation is less likely to compromise the quality or sustainability of OpenReview as a resource.

We call upon researchers, conference organizers, and practitioners—particularly those working at the intersection of machine learning and language models—to jointly define, refine, and adopt such benchmarks. This collaborative process must also engage with broader ethical considerations, including the protection of author and reviewer privacy, responsible anonymization of sensitive content, and the mitigation of representational biases. For example, research areas with more abundant data may inadvertently dominate the training signal, potentially skewing the learned priorities of evaluation models.

**Ultimately, the continued value of OpenReview as a shared academic asset depends on proactive, collective stewardship by the community it serves.**

# 7 Alternative Views

**LLM-based Review and Research.** Some may argue that LLMs are becoming more and more capable of finishing scientific research and evaluation, and should eventually replace human reviewers. If models can predict review scores or generate critiques that approximate expert judgment, then preserving human oversight might appear unnecessary or even inefficient. **Our view**: We argue that peer review is not just a filtering mechanism, but a deliberative process that helps shape scientific values and standards [150, 151]. Over-reliance on automation risks eroding its collaborative and interpretive nature [152, 153]. LLMs, while powerful, are not reliable in reasoning with the same contextual grounding or responsibility as human experts [154, 155]. Human reviewers must remain responsible for interpreting and controlling LLM tools [154]. Interactions between authors and reviewers should stay dialogic and grounded in fairness, not reduced to rigid or opaque evaluations [152]. Moreover, systems must guard against hallucination, adversarial misuse, and bias propagation [156–158]. Evaluation frameworks built on OpenReview should align with scientific values rather than model evaluation metrics [22, 106].

**Inconsistency of Peer Review Limits Its Usefulness for Alignment.** One concern is that peer review data may be too noisy or inconsistent to serve as a reliable supervision signal [159]. Reviewers often disagree on paper quality, assign divergent scores, or emphasize different aspects of a submission. Given this subjectivity, it may be argued that using such data for alignment could reinforce inconsistent or unstable behaviors in LLMs. **Our view**: Rather than aiming for deterministic consensus, alignment in this context involves modeling disagreement, grounding claims, and reasoning for underlying conflicts. This perspective is increasingly emphasized in recent alignment literature [68, 70]

**Scientific Review Tasks May be Too Narrow to Generalize.** Another possible objection is that scientific reviewing and paper writing are narrow, domain-specific tasks that do not generalize to broader LLM capabilities. Models trained on OpenReview may excel at research-related tasks but fail to transfer to everyday use cases, limiting their value as general-purpose assistants. **Our view**: Research tasks serve as high-complexity instances of structured human reasoning, with grounded stakes and verifiable outcomes. Learning from these tasks offers not only domain expertise but also training in core cognitive patterns that generalize across domains. Recent success of DeepResearch-type of products [8] explicitly aim to generalize research workflows into agentic LLM behaviors.

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

## A Long-lasting effect of low-quality reviews

Here, we provide a simplistic population genetic model to capture our intuition that a fast-growing reviewer body's lack of training can have a long-lasting effect even after the field matures by setting up precedent. Note that this is an extremely simplified model, and we acknowledge that reviewer quality is not binary and can depend on many factors.

We follow the standard Wright-Fisher model in population genetics. For each review round $t$, there are $G_t$ "good" reviews and $B_t$ bad reviews (in total $N_t = G_t + B_t$ reviews). In generation $t + 1$, for $N_{t+1}$ new reviews, we model them as generated randomly, with some level of preference. Formally

$$B_{t+1} \sim \text{Binomial}\left(N_{t+1}, \frac{B_t}{(1 + s(t))G_t + B_t}\right) \tag{1}$$

where $s(t)$ is a factor for preference that could change over time, ideally $s(t) > 0$ so that one has a preference towards writing less bad reviews than simply replicating what was seen in the past cycle. We define $X_t = \frac{B_t}{N_t}$.

We can take a diffusion limit of the model, and the proportion of bad reviews can be approximated as a Wright-Fisher SDE

$$dX_t = s(t)X_t(1 - X_t)dt + \sqrt{\frac{X_t(1 - X_t)}{N(t)}}dW_t \tag{2}$$

where $W_t$ is a one-dimensional Brownian motion.

We numerically solve the corresponding Fokker-Planck equation for different $N(t)$ and intervention $s(t)$. We assume that $N(t)$ follow a logistic growth representing the usual maturing of the field. The results are given in Fig.3. **The takeaway message is that we need to act early in stopping the trend of preferring low-quality reviews to prevent the downgrade of overall quality and setup the precedent for the next generation to follow.** The trend can still be reversed in a mid to late stage, but requires more efforts (cf. first and last row in Fig.3, we need a longer period of selection if we started late). It is useful even just to stop, instead of reverting, the current trend of preferring bad reviews (cf. second row of Fig.3). The intuition behind these results is that if a once rare bad review was fixed into the norm during the expansion of the field, it will be part of the norm and hard to be filtered out in the future when the field grows even larger.

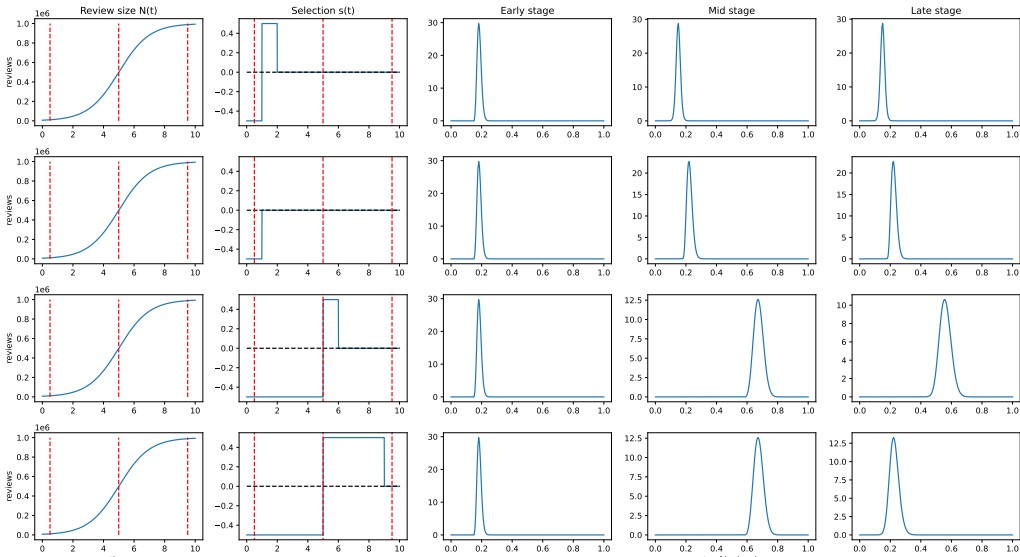

Figure 3: Distribution of frequency of bad reviews under Wright-Fisher type of selection model. The three stages of time are marked in red vertical lines in the first two panels. First column: model number of reviews, Second: what selection we put at which time, Third-last: distribution of proportion of bad reviews.

