# OpenReview forum: "OpenReview Should be Protected and Leveraged as a Community Asset for Research in the Era of LLMs"
_NeurIPS.cc/2025/Position_Paper_Track — Submitted to NeurIPS 2025 Position Paper Track_

### Official Review · Reviewer_ixFy · 2025-07-05

**Significance:** 2
**Presentation:** 3
**Rating:** 5
**Confidence:** 3

**Summary:**

This paper argues that the ML, AI community should make more use of OpenReview assets (reviews, discussions, ideations) to research LLM (and agents). The assets can be used for (1) peer reviewing, where LLMs are trained to produce better reviews and as such assist human reviewers (2) writing better research papers assisting authors to investigate new research topics, (3) aligning LLMs for better reasoning. As OpenReview contains multi-turn conversations and interactions between authors and reviewers, the authors believe that OpenReview asset will have the potential to enhance researches in related fields of LLMs (Agentic tasks, Open-Ended research).

**Strengths:**

-	Clear formatting, easy to follow, supported with evidences for each arguments.
-	Have addressed future work opportunities for each of the points raised.
-	Personally, the reviewer disagrees with the current position (which means that this paper will spark discussions and is worth being highlighted as a position paper). Fine-tuning LLMs for tasks relevant to OpenReview can potentially encourage the misuse of LLMs by reviewers in current review practices. As such, rather than promoting OpenReview for benchmarks, OpenReview should ban the utilization of discussion data for benchmarks. Would be happy to hear the author’s thought on this.

**Weaknesses:**

-	While OpenReview contains a large number of discussions, the data is mostly limited to ICLR conferences. Other conferences only have 1-2 years of history of releasing author-review discussions. As such, the constructed task and dataset might be extremely narrow. Have the authors taken this into consideration? How does the datasets listed in Table 1 and the proposed dataset devoid from this problem?
-	The reviewer is also concerned that creating and promoting such tasks from the OpenReview will only accelerate the current misuse of LLMs for reviewing (using LLMs to generate reviews). It would be nice to enforce these discussions in 7.Alternative Views first section.
-	It might be helpful to consider and discuss the below papers which seems quite relevant to this work.

[1] Mapping the increasing use of llms in scientific papers

[2] Position: The AI conference peer review crisis demands author feedback and reviewer rewards

**Questions:**

-	The current status quo of OpenReview assets has not been thoroughly discussed in this paper. While Table 1 provides a list of tasks related to OpenReview, it is simply a list of works. A more thorough analysis of the current tasks and how the new tasks should be set should be more specific in details.
-	Why can't existing datasets be used for the arguments made in the paper? What is the major bottleneck to utilizing these assets as a training dataset for LLMs?
-	The reviewer has personally collected ICLR review datasets for experiments. One of the major problems was the inconsistency in peer review questions and criteria every year. As such, the structure is not unified throughout the years. How can we handle such practical problems?
-	The reviewer is slightly confused about why this paper was submitted as a position paper, but not to a dataset & benchmark track. What is stopping the authors from creating such a dataset?

**Alternative Position:**

Yes, and alternative positions are well-considered and named but not addressed

**Author Identification:**

No.

**Context:**

3

**Discussion:**

3

**Ethics:**

["NO or VERY MINOR ethics concerns only"]

**Position:**

Yes, the paper argues for or against a position related to machine learning.

**Support:**

2

**Thoroughness:**

3

---

### Official Review · Reviewer_y5eR · 2025-08-08

**Significance:** 3
**Presentation:** 3
**Rating:** 5
**Confidence:** 4

**Summary:**

This position paper argues that OpenReview as a continually evolving repository of peer reviews, author rebuttals, meta-reviews, and decision outcomes should be leveraged as a core community asset for advancing LLM research. The authors highlight three key application areas: (1) enhancing peer review quality, scalability, and accountability through automated assistance and protection tools; (2) enabling open-ended benchmarks rooted in genuine expert deliberation to support model evaluation and post-training; and (3) providing high quality datasets for alignment research that capture real-world expert interactions, intentions, and scientific values. They propose collaborative exploration of standardized benchmarks, ethical guidelines, and usage protocols to ensure responsible and beneficial use of OpenReview data. The central position is that, with proper safeguards and stewardship, OpenReview can uniquely advance LLM capabilities while strengthening the research community.

**Strengths:**

The paper’s strongest contribution is its articulation of three concrete use cases for leveraging OpenReview in LLM workflows: assisting and protecting peer review, enabling open-ended task evaluation and post-training, and creating a high quality dataset for alignment and reasoning. The peer review assistance case is well developed, listing specific automated tasks such as review drafting, rebuttal mediation, calibration checks, and meta-review generation, with opportunities that are actionable and relevant to current conference workflows. The open-ended task evaluation section clearly positions OpenReview as a rich, corpus driven by experts and identifies distinct supervision streams, while the alignment and reasoning case compellingly frames multi-round peer review as a realistic setting for deliberative reasoning and pluralistic value alignment. These use cases show a practical vision for bridging peer review infrastructure and LLM development using OpenReview.

**Weaknesses:**

The three proposed use cases are unevenly developed. Peer-review assistance is described with specific example tasks, but lacks concrete deployment plans, governance structures, or evidence of effectiveness. The open-ended evaluation and alignment applications are presented largely at a conceptual level, with minimal detail on benchmarking protocols, success metrics, or mechanisms to address bias, data leakage, and disagreement modeling. Across all cases, the paper provides limited discussion of privacy protections, consent processes, and ethical safeguards, which are critical when handling sensitive review data. No pilot studies, simulations, or feasibility analyses are offered to assess practicality, cost, or integration with existing OpenReview workflows. Without clearer implementation pathways and risk mitigation strategies, the proposals remain speculative, leaving uncertainty about how they would be operationalized or adopted by the community.

**Questions:**

How do you envision addressing privacy, consent, and governance concerns when using sensitive review data for LLM training and evaluation, particularly in light of potential reviewer identification risks?
How would you mitigate the risk of bias or overfitting in LLMs trained on OpenReview data, given the narrow domain and potential systemic patterns in reviewer judgments?

**Alternative Position:**

Yes, and alternative positions are well-considered and named but not addressed

**Author Identification:**

No.

**Context:**

3

**Discussion:**

3

**Ethics:**

["NO or VERY MINOR ethics concerns only"]

**Position:**

Yes, the paper argues for or against a position related to machine learning.

**Support:**

3

**Thoroughness:**

4

---

### Official Review · Reviewer_QPFF · 2025-08-12

**Significance:** 3
**Presentation:** 3
**Rating:** 5
**Confidence:** 3

**Summary:**

The paper argues that OpenReview should be treated as a shared, protected community asset for research in the LLM era. It outlines how OpenReview data could enhance peer review quality and scalability, provide expert-grounded benchmarks for open-ended LLM tasks, and serve as a rich source for alignment and reasoning research. It also addresses structural risks to review quality from rapid conference growth and calls for standardized benchmarks and responsible stewardship.

**Strengths:**

- Clearly identifies multiple concrete use cases (peer review enhancement, open-ended LLM evaluation, alignment research) with detailed examples of possible applications.
- Highlights the unique value of OpenReview data (expert-authored, multi-round, domain-specific) compared to synthetic or crowd-sourced datasets.
- Incorporates discussion of ethical considerations, alternative perspectives, and risks of over-automation in peer review.

**Weaknesses:**

- The proposed actions (e.g., benchmark creation, stewardship mechanisms) remain high-level without concrete implementation roadmaps or prioritization.
- Limited empirical evidence or case studies demonstrating feasibility of the proposed LLM-assisted peer review improvements.
- Risks of bias and inconsistency in peer review data are acknowledged but not deeply addressed in terms of mitigation strategies beyond general “quality protection.”
- The openreview data could be biased, subjective, and even noisy, which should be addressed.

**Questions:**

na

**Alternative Position:**

Yes, and alternative positions are well-considered and addressed by the argument

**Author Identification:**

No.

**Context:**

3

**Discussion:**

3

**Ethics:**

["NO or VERY MINOR ethics concerns only"]

**Position:**

Yes, the paper argues for or against a position related to machine learning.

**Support:**

2

**Thoroughness:**

3

---

### Note · Authors · 2025-08-26

**1-10 Additional Comments:**

One reviewer raised concerns about the availability and scope of the OpenReview dataset. We would like to clarify that, as a position paper, our goal is not to release a finalized dataset but to highlight the importance of OpenReview as a shared community asset and to encourage broader discussions on its responsible use.

While ICLR is currently the only venue with fully open peer-review records, its multi-year dataset already provides a valuable foundation for exploratory experiments and controlled investigations, where all reviews remain strictly anonymized.

Looking ahead, we believe that demonstrating the value of responsible, verified techniques on this existing dataset will generate stronger evidence to convince both the community and other conferences to contribute more review data in a principled, collaborative manner. This gradual, evidence-driven process is precisely why we believe the position argued in our paper is timely and important.

**1-11 Submit Again:**

Probably yes

**1-1 Submission Process:**

4

**1-2 Next Year:**

We would appreciate it if the position paper track could provide a more transparent timeline, clearer review guidelines, and a better-documented review process next year. This would help authors better understand the expectations and improve the quality and relevance of submissions.

**1-3 Future Development:**

1. **Open discussion forum**: It would be valuable to provide a dedicated forum open to the public where authors, reviewers, and the broader community can discuss the positions presented in the track. This could facilitate broader engagement and constructive debate around emerging viewpoints.

2. **Follow-up study or report track**: Consider introducing a track where authors of accepted position papers from previous years can provide updates on progress in the field. Such follow-ups, over a period of 2–5 years, could lead to a series of survey-style reports in highly active or rapidly developing areas.

3. **Evaluation of impact and evolving perspectives**: The quality and outcomes of these follow-up discussions and reports could serve as one way to evaluate the impact of position papers. They could also reflect how deeply authors (and the broader community) believe in their originally argued positions — or how perspectives may evolve over time as the field progresses.

**1-4 Interest:**

["Panel discussions with other position paper authors", "Structured debates on controversial topics", "Workshops for developing position papers", "Mentorship programs for early-career researchers", "Other (please specify in the next question)"]

**1-4 Other Interest:**

We would be particularly interested in contributing to initiatives that improve reviewer training and alignment with review guidelines to elevate reviewing and discussion quality -- not only on OpenReview (as discussed in our position paper) but also across other ML conferences.

Consistent with our position paper, these efforts can be responsibly supported by human(e.g., senior researchers/AC/SAC)-in-the-loop, AI-assisted tools. We’d be glad to help design or participate in pilot programs, training modules, and evaluation studies that advance this broader community vision.

**1-5 Thoughtful:**

8

**1-6 Supportive:**

8

**1-7 Technical Aspects Versus Position:**

5

**1-8 Gate Keeping:**

8

**1-9 Camera Ready Changes:**

We thank the reviewers for their thoughtful and constructive feedback.
Based on their suggestions, we will clarify the following points in the camera-ready version to further strengthen our position.

**To foster constructive discussion, we will also include representative questions and diverse viewpoints raised by the reviewers, together with our responses, in the camera-ready revision**:

- Clarifying the scope and objective: We will emphasize that this paper advocates for responsible stewardship of OpenReview data and calls for community-driven efforts to explore benchmarks, governance models, and best practices, rather than releasing a finalized dataset or benchmark.

- Highlighting and contrasting risks: As unregulated LLM usage in peer-review systems is becoming a concern, we will explicitly contrast current undesirable practices with our vision of responsible, human-in-the-loop integration of LLMs for assisting and improving the peer-review process.

- Integrating relevant recent work: We will incorporate discussions of recent papers and concurrent preprints suggested by reviewers to further situate our position within ongoing community debates.


We believe these clarifications strengthen our position and better highlight the paper’s role as a **starting point for coordinated community action**, which is fully aligned with the goals of the **NeurIPS Position Paper Track**.

**3-1 Review Response1:**

QPFF

**3-2 Reaction To Review1:**

We thank the reviewer for their thoughtful feedback and constructive suggestions. We are glad that the reviewer finds the proposed use cases of OpenReview data valuable and timely. We address the main concerns below.

- 1. On concrete implementation and actions

We would like to clarify that, as a position track paper, our primary goal is not to present finalized solutions but to highlight the strategic value of OpenReview data and initiate a community-wide discussion around benchmarks, stewardship, and governance. Nevertheless, we will improve the position in the camera-ready version by sketching possible directions for benchmarks and stewardship mechanisms, while making clear that these are starting points rather than fixed designs.

- 2. On feasibility and empirical evidence of LLM-assisted peer review

While we intentionally avoid overcommitting to specific implementations, we will strengthen the connection to recent pilot initiatives such as ICLR 2025’s Review Feedback Agent, NeurIPS 2024’s Checklist Assistant, and AAAI 2025 AI-assisted multi-phase review, which illustrate the practicality of LLM-assisted review support and responsible evaluation.

- 3. On bias, subjectivity, and noise

We agree that this is an important issue and will expand our discussion in Section 3. Rather than assuming OpenReview data is clean, we will emphasize the need for community-driven quality assurance, including reviewer calibration, rubric normalization, and pluralistic evaluation methods that treat disagreement as a signal rather than noise.

Overall, we believe these clarifications will make the paper’s position clearer: our contribution is to frame an urgent agenda and catalyze collective action, not to propose a finalized technical solution.

**3-3 Review Response2:**

y5eR

**3-4 Reaction To Review2:**

We thank the reviewer for their thoughtful feedback and constructive suggestions. We appreciate the recognition of the value and timeliness of leveraging OpenReview data and address the main concerns below.

- 1. Uneven development across the 3 use cases

We acknowledge that the peer-review assistance case is described in greater detail. This is **intentional**: ensuring **data quality and reviewer calibration** is a foundation for all subsequent applications, including open-ended evaluation and alignment reasoning. Since these latter directions **build upon reliable review data**, we keep their discussion at a higher level of abstraction to focus on highlighting the opportunities and challenges.

As a **position paper**, our goal is to **prioritize foundational issues first** and provide a perspective rather than finalized solutions, consistent with the NeurIPS CFP, which states that *“position papers argue for what should be done rather than report completed advances.”* In the camera-ready version, we will enrich Sections 4 and 5 with illustrative examples to clarify potential directions for benchmarks and evaluation tasks.

- 2. On governance, privacy, and consent

We greatly appreciate the reviewer’s thoughtful concerns, which confirm that this topic is of genuine discussion value—the core purpose of our position paper. Rather than aiming to **solve all challenges immediately**, we hope to **activate broader community discussion** on governance, privacy, and consent, and to **work collaboratively with conference organizers, OpenReview maintainers, and ethics advisors** toward principled solutions. In the camera-ready version, we will clarify this scope and emphasize that our contribution is to **catalyze these conversations**.


**Overall**
We believe these clarifications strengthen the position and highlight the paper’s role as a **starting point for collective community action**, fully aligned with the goals of the **NeurIPS Position Paper Track**.

**3-5 Review Response3:**

ixFy

**3-6 Reaction To Review3:**

We thank the reviewer for their thoughtful and constructive feedback. We appreciate that, while perspectives differ, the reviewer recognizes this paper’s potential to encourage valuable discussion within the community. We believe that surfacing diverse viewpoints is essential for shaping responsible practices around OpenReview data, and we appreciate the opportunity to clarify our position below.

- 1. Scope of OpenReview

We intentionally focus on ICLR, which currently provides the most comprehensive and continuous open-review dataset. Starting with this venue allows us to manage potential risks and develop responsible practices before expanding to other conferences.

Please also see *1-10 Additional Comments*.

- 2. Rubric inconsistency and review criteria

We agree that rubric inconsistency across years and venues presents a technical challenge. Addressing this requires **community-wide collaboration**, which reinforces the motivation behind our position paper. Future standardization efforts can make OpenReview data **more reliable and useful** for the entire community.

- 3. Position paper rather than a dataset paper

This paper’s goal is **not to construct a dataset or resolve all technical details**, but to **encourage community-wide discussion** on the responsible stewardship and use of OpenReview data. As a **shared community asset**, we believe that its sustainable protection and long-term value depend on **collaborative governance** rather than isolated/individual solutions.

- 4. Discussion of relevant papers

We thank the reviewer for pointing out concurrent related works (note the position paper mentioned is arXiv'd in May 2025), and we are happy to add discussion in our revision.

**Overall**

We hope these clarifications bring attention to the fact that the reviewer’s concerns reflect the **open challenges** our paper aims to discuss, which aligns closely with the goals of the **NeurIPS Position Paper Track**.

---

### Meta-Review · Area_Chair_TDQJ · 2025-09-18

**Rating:** 5
**Confidence:** 3

**Strengths:**

The paper argues for the use of OpenReview data as resources for LLM research and presents three specific use cases. The ideas are well-thought and clearly-presented. Openreivew serves as the most publicly available review data available - there are many potentials in the era of LLM.

**Weaknesses:**

The paper did not discuss the ethic aspects of the use of OpenReview data, such as copyright, privacy, etc. These are really important issues that need to be addressed and discussed, especially for most cases, authors and reviewers were not aware of how the data would be used when submitting papers to openreview.

**Questions:**

Will the authors/reviewers have an opt-out option?

**Ethics:**

There are ethical violations or concerns raised by reviewers.

**Thoroughness:**

3

---

### Decision · Program_Chairs · 2025-09-26

Reject